



# Austrian NIR Soil Spectral Library for Soil Health Assessments

Julia Fohrafellner[1], Maximilian Lippl[2], Armin Bajraktarevic[1], Andreas Baumgarten[1], Heide Spiegel[1], Robert Körner[1], Taru Sandén[1]

[1]Department for Soil Health and Plant Nutrition, Austrian Agency for Health and Food Safety (AGES), Vienna, 1220, Austria
[2]Department for Feed Analysis and Quality Testing, Austrian Agency for Health and Food Safety (AGES), Vienna, 1220, Austria

*Correspondence to*: Julia Fohrafellner (julia.fohrafellner@ages.at)

**Keywords:** dataset; EJP SOIL; near-infrared; ProbeField; spectroscopy

**Short summary**

The first openly accessible Austrian near-infrared (NIR) Soil Spectral Library was developed, including over 2100 samples covering all Austrian environmental zones. The prediction of soil properties via partial least square regression showed

potential, but the accuracy was insufficient compared to routine laboratory analyses. We encourage using the open Library as a foundation for further spectral analysis and modelling and we support future soil health assessments via spectroscopy.

**Abstract**

The rise in demand for soil data and information calls for quick and cost-effective methodologies to quantify soil properties.

This is particularly important in the realm of restoring soil health in Europe. Near-infrared (NIR) spectroscopy has demonstrated the ability to predict specific soil properties with high accuracy whilst being less costly and time-consuming than traditional methods. To fill gaps in national spectroscopic soil data, we compiled the first Austrian NIR Soil Spectral Library (680–2500 nm) based on legacy samples (n = 2129), covering all environmental zones of Austria. We then applied partial least square regression modelling to test the usability of the dataset for soil health assessments at its current stage. Our analysis

revealed that, at the present time, the Austrian NIR Soil Spectral Library is not suitable to predict most of the 14 soil properties with sufficient accuracy. Nevertheless, total nitrogen, $CaCO_3$, soil organic carbon and clay showed satisfactory results ($R^2 >$ 0.7). Most importantly, the dataset containing sample meta-data (e.g., land use type, environmental zone or zip code), laboratory reference values and NIR spectra with 1 nm resolution can be used as a foundation for further spectral analysis and modelling. We make this work openly accessible to actively contribute to closing soil data gaps and promote the expansion of

soil spectral libraries as a basis for soil health assessments.



## 1 Introduction

In 2021, the European Commission published the renewed Soil Strategy (European Commission, 2021), in which the vision for healthy soil ecosystems by 2030 was presented. Soil health, defined as "the continued capacity of soils to provide ecological functions for all forms of life, in line with the Sustainable Development Goals and the Green Deal" (European Commission et al., 2020), is still an evolving term and concept. It received heightened attention since the implementation of the EU Mission: A Soil Deal for Europe (European Commission et al., 2020) and particularly the discussed Soil Monitoring Law (European Commission, 2023). Narrowing down the definition of soil health and, most importantly, quantifying it has therefore become a key issue for soil scientists. Characterizing soil health in the form of multiple indicators and creating a "soil health index" (Lehmann et al., 2020) suitable to measure and monitor soil health in the EU (European Commission, 2023; Matson et al., 2024) are pivotal in achieving healthy soils until 2030. Efforts to define such an index are multi-fold (Matson et al., 2024; Wade et al., 2022; Rinot et al., 2019), but generally they agree on including chemical, physical and biological indicators that go beyond crop production to encompass soil ecosystem services (Shen and Teng, 2023). Properties such as soil organic carbon (SOC), soil nutrients, pH or cation-exchange capacity (CEC) are frequently included in soil health assessments (Lehmann et al., 2020). These properties are routinely analyzed in laboratories with traditional methods but are often time-consuming and require resources such as expensive equipment and chemicals. Based on the increasing requirements for soil health assessments and monitoring posed by the Soil Monitoring Law, less cost-intensive alternative methods are in demand (Safanelli et al., 2025).

The application of soil visible (vis) and near infrared (NIR) spectroscopy to predict soil properties, particularly chemical ones, using statistical and machine learning methods has increased rapidly in recent decades (Viscarra Rossel et al., 2011; Gholizadeh et al., 2013; Stenberg et al., 2010). Spectroscopy was shown to generate fairly to very accurate estimates of e.g., total carbon (Ma et al., 2023), SOC (Guerrero et al., 2016) and its fractions (Jaconi et al., 2019b), as well as soil texture, specifically clay (Jaconi et al., 2019a), and carbonates (Tavakoli et al., 2023). This approach also bears great potential for soil fertility assessments of total nitrogen (Park et al., 2024) along with total and critical available phosphorus (Recena et al., 2019). Compared to traditional laboratory analyses, spectroscopy has many advantages because it is fast, simple, cost-effective, reproducible, repeatable, non-destructive and environmentally friendly (Viscarra Rossel et al., 2006; Nocita et al., 2015; Soriano-Disla et al., 2013). To improve predictions and fill data gaps, large reference training data sets, so-called soil spectral libraries (SSL), are being built and often made freely available. These extend from local, regional to national and even global scale (Fao, 2022), thereby helping to describe soils and their health whilst improving soil data availability (Cornu et al., 2023). Nevertheless, large areas without available data in the global coverage of NIR SSL remain, calling for active participation to fill these gaps (Viscarra Rossel et al., 2016; Safanelli et al., 2025). This is necessary because the predictive capacity of models relies on the number and diversity of soil samples and conditions represented in the spectra. Concurrently, new methods for coordinating existing SSL are being developed, enabling interoperability between labs, data harmonization, engagement of communities and model development (Safanelli et al., 2025; Peng et al., 2025).



The application of NIR spectroscopy for Austrian soils is limited so far. Beyond sample analysis within the LUCAS inventory (Fernandez Ugalde et al., 2022), Ludwig et al. (2023) tested the suitability of vis-NIR and MIR (mid infrared) spectroscopy for forest soils whilst comparing different modelling approaches. Moreover, the suitability of vis-NIR for measuring soil carbon contents and the effects of agricultural management methods was analyzed by Bieber (2023) on a regional scale. To date, no national SSL for Austria is openly available. By using available legacy soil samples, including their results from chemical and physical analysis, we wanted to fill this gap and analyzed them via NIR spectroscopy. Therefore, the objectives of this study were (i) to provide a first dataset on Austrian NIR soil spectra and reference laboratory analysis for several soil health properties, covering all Austrian environmental zones and (ii) to apply partial least square regression (PLSR) for model calibration and validation and test the model's applicability for national soil health assessments.

## 2 Soil sample selection

The selection of soil samples for the Austrian NIR Soil Spectral Library was based on a wide distribution of soils with different properties to strengthen the dataset usability and predictive power of applied models. The dataset includes legacy soil samples (analyzed and stored in the AGES archive) from several long-term field experiments, different project campaigns and from farmers' land. Moreover, samples from the so-called AGES "soil box" (Ages, 2025), which are sent in by private persons, were included. The soil box allows individuals to have their soils analyzed for common properties such as pH, SOC and phosphorus. These samples are special because they stem from a wide range of land uses including grassland, arable land, forests, orchards, hedges or lawn, but also compost or garden soils. This particularly promotes the diversity of included soils and enhances the geographical distribution and coverage throughout Austria. All included samples were collected between 2016 and 2023 and soil sampling depth ranged between 0 cm down to a maximum of 110 cm (mean: 24.5 cm). This yielded a total of 2129 samples which were considered for the soil spectral library, covering all environmental zones of Austria (Fig. 1). Most samples represent the Pannonian zone (n = 1059), followed by the Continental (n = 778), Alpine South (n = 286) and Mediterranean Mountains (n = 5) zones. For one sample, the location and environmental zone are unknown (Sample_number 743).

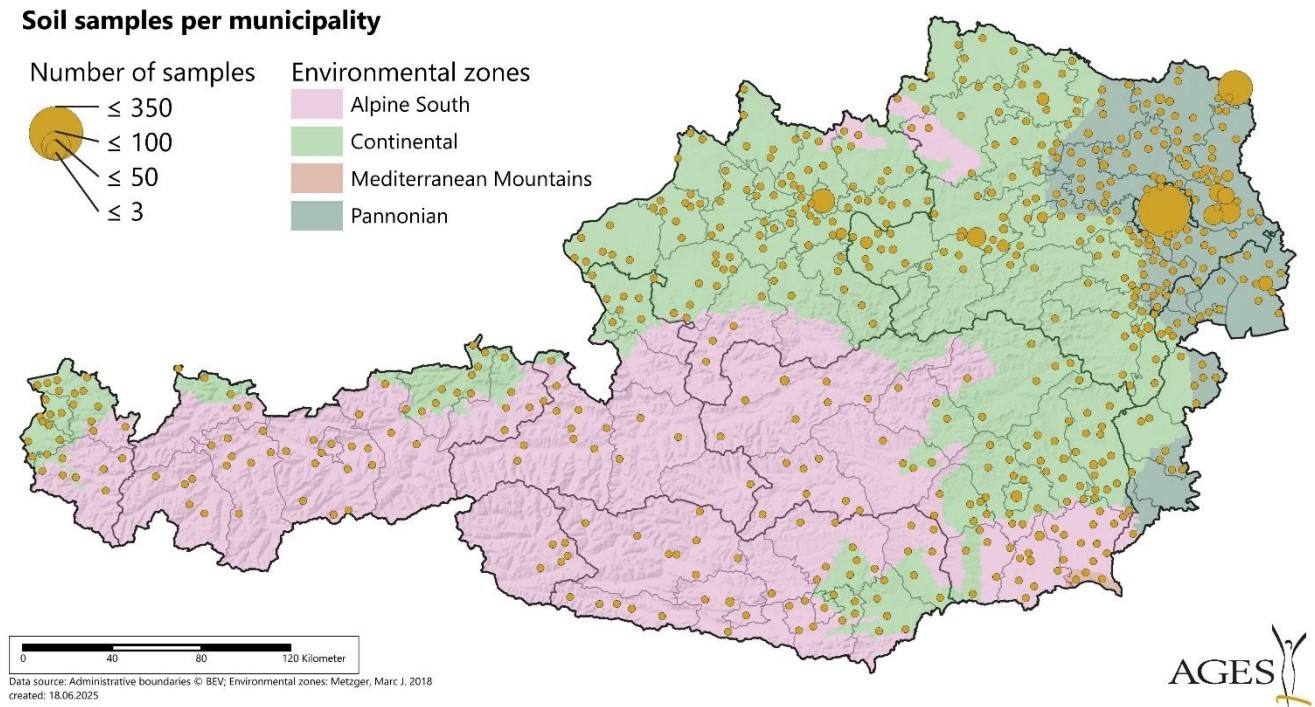

**Figure 1: The environmental zones in Austria according to Metzger et al. (2018; 2005) with sampling locations of the NIR spectral library (n = 2128) sampled between 2016 and 2023. Circle size indicates the number of soil samples per municipality.**

**3 Dataset creation and description**

Information about the soil samples included in the dataset was extracted from the AGES internal database "LISA". Missing data regarding zip codes and land-use type was gathered by contacting responsible persons within and outside of AGES. MS Excel© (version 2410) and the statistical programming language R within the RStudio interface (version 2024.04.0) were used for data processing. The dataset contains 1846 columns, of which columns 1 to 11 provide information on the soil sample,

columns 12 to 25 contain chemical and physical reference analysis results, and columns 26 to 1846 contain the measured absorbance from 680 nm to 2500 nm for every nm. Column "Sample_number" assigns each sample a number, starting from one, and column "Sampling_year" shows the year of soil sampling. In column "Experiment_number" the samples were grouped into experiments which indicate that samples were collected on the same site. When several samples were sent in together but originated from different sites (e.g., many sites sampled within one research project), this was indicated by an

underscore (e.g., 9_1, 9_2). Column "Sample_source" provides information on the source of the soil sample, e.g. whether it was taken from a long-term experiment (LTE), part of a research project sampling campaign, or submitted by a private person or advisor for analysis. The columns "Zip_code" and "Municipality_code" indicate the location of the sampling site, which in some cases needed to be approximated due to limited location information. Providing coordinates was not possible because



the dataset includes samples sent in by private individuals and thereby classifies them as sensitive data. The environmental
zone (Metzger et al., 2005; Metzger, 2018) of the sampling location is shown in column "Environmental_zone". Sampling
depths are reported in columns 8 and 9, land use types in column 10. Most samples were taken from arable land (n = 1485),
followed by vegetable plots (n = 202) and lawn (n = 174). Other land use types (orchards, forests, grassland, hedges, ornamental
plants and vineyards) were sampled less than 100 times. For 73 samples the land use type was unknown. Lastly, column 11
indicates whether samples were sent in for analysis with a "soil box". As these samples are taken and submitted by private
individuals, the land use type defined by those individuals might not always be accurate. Finally, such samples may contain
artificial substrate, high amounts of compost or fertilizers. Regarding the chemical and physical reference analysis and NIR
spectroscopy, the methodologies and results are described in detail in the following sections. The dataset is available at
https://zenodo.org/ in the form of an excel file, which is accompanied by document metadata and a legend (Fohrafellner et al.,
2025).

## 4 Analysis of soil

### 4.1 Chemical and physical reference analysis

Chemical and physical soil properties were analyzed in the AGES soil laboratory for all soil samples (Table 1). The samples
were air-dried at maximum 40°C for at least two days and sieved through a 2 mm stainless steel sieve (ÖNORM L 1053:2012-
04-15). Soil organic carbon (SOC) and total carbon (TC) were analyzed by dry combustion in a LECO TruMac CN (LECO
Corp., St. Joseph, MI, United States) at 650°C and 1250°C, respectively (ÖNORM L 1080:2013-03-15). The labile carbon
was determined according to Tatzber et al. (2015). The carbonate content ($CaCO_3$) was measured gas-volumetrically according
to the "Scheibler method" ($CO_2$ evolution; ÖNORM L 1084:2016-07-01). Total N (TN) was determined via elemental analysis
using a LECO Trumac CN at 1250° C (ÖNORM EN 16168:2012-10-01). Plant available phosphorus (hereafter referred to as
"Phosphorus") was determined by calcium-acetate-lactate (CAL) extraction (ÖNORM L 1087:2012-12-01). Soil pH was
measured electrochemically (pH/mV Pocket Meter pH 340i, WTW, Weilheim, Germany) in 0.01 M $CaCl_2$ at a soil-to-solution
ratio of 1:5 (ÖNORM EN 15933:2012-10-01) and in calcium acetate (Vdlufa, 1991) to compare the suitability of NIRS to
predict the results of these methods. The cation exchange capacity (CEC) was determined by extracting the effective
exchangeable cations $Ca^{++}$, $K^+$, $Mg^{++}$, $Na^+$ and $Al^{+++}$, $Fe^{+++}$, $Mn^{++}$ and $H^+$ in barium chloride solution (ÖNORM L 1086-1:2014-
03-15). Texture was determined according to ÖNORM L 1061-2:2002-02-01, and clay was further analyzed by density in soil
suspension (ÖNORM EN ISO 17892-4).

The models for calibration and validation (section 5 "Spectroscopic modelling") for each soil property were tested with the
full range of SOC values, but also with subsets containing only samples with SOC values below 7%, as this is the upper limit
for most agricultural mineral soils in Austria. The modelling results revealed that using only the latter subsets generated better
estimates the properties SOC, SOC/clay ratio and TC. We therefore provide the summary statistics for these properties using
all samples or only samples with less than 7% SOC (Table 1).



**Table 1: Summary statistics of the measured soil properties of the Austrian NIR Soil Spectral Library for all samples and for the subset of samples with SOC below 7%. SOC is soil organic carbon, TC is total carbon, TN is total nitrogen, and CEC is the cation exchange capacity.**

| Property | $n$ | Min | Max | Median | Mean | SD | Skewness |
|---|---|---|---|---|---|---|---|
| **All samples** | | | | | | | |
| SOC (%) | 2112 | 0.02 | 44.8 | 1.96 | 2.76 | 3.20 | 6.45 |
| SOC/clay ratio | 534 | 0.0009 | 1.59 | 0.103 | 0.199 | 0.241 | 2.37 |
| TC (%) | 92 | 1 | 39.1 | 4.90 | 6.54 | 6.23 | 2.79 |
| Labile carbon (mg kg$^{-1}$) | 567 | 47 | 1516 | 558 | 600 | 194 | 1.57 |
| CaCO3 (%) | 327 | 0.01 | 81.9 | 9.7 | 13.3 | 12.6 | 1.90 |
| TN (%) | 1036 | 0.029 | 2.42 | 0.192 | 0.253 | 0.206 | 3.95 |
| Phosphorus (mg kg$^{-1}$) | 1643 | 1 | 1624 | 78 | 113 | 137 | 4.20 |
| pH (CaCl2) | 1917 | 3.18 | 7.93 | 7.26 | 6.92 | 0.848 | -1.58 |
| pH (Acetate) | 304 | 5.59 | 7.42 | 6.48 | 6.53 | 0.391 | 0.29 |
| CEC (cmolc kg$^{-1}$) | 641 | 2.88 | 85.4 | 21.0 | 20.9 | 8.75 | 2.33 |
| Sand (%) | 562 | 5.6 | 92.5 | 32.2 | 36.2 | 18.7 | 0.59 |
| Silt (%) | 562 | 5 | 75.7 | 47.1 | 45.7 | 14.3 | - 0.37 |
| Clay (%) | 534 | 1.5 | 47.1 | 17.3 | 18 | 9.05 | 0.50 |
| Clay in suspension (%) | 381 | 10 | 40 | 20 | 21.4 | 5.19 | 0.77 |
| **SOC < 7%** | | | | | | | |
| SOC (%) | 1997 | 0.02 | 6.94 | 1.91 | 2.20 | 1.17 | 1.60 |
| SOC/clay ratio | 518 | 0.0009 | 1.25 | 0.0978 | 0.182 | 0.214 | 2.37 |
| TC (%) | 71 | 1 | 8.82 | 3.93 | 3.92 | 1.68 | 0.229 |


## 4.2 Spectral measurement and preprocessing

The legacy soil samples from the AGES soil archive were prepared as for previous reference analysis and measured with the SpectraStar™ XL near-infrared spectrometer from Unity Scientific (Fig. 2) between the range 680 to 2500 nm. This instrument was equipped with 2 detectors and provided a spectral resolution of 1.0 nm, with the sensor switch being between 1340 and

1341 nm. A matte-surface gold-plated metal reflector was used as an internal reference. The sample cup (8.5 cm diameter) was filled up to about one third with a soil sample and covered with the reflective lid, which was pressed slightly onto the sample. Each sample was scanned 24 times, and a mean was calculated. The same soil sample was then transferred to a new sample cup and the scanning was repeated. A mean was calculated from these two repetitions. The standard used was White Sand (Lucky Bay, Australia), which enables harmonizing results from different near-infrared spectrometers used within the

ProbeField project.





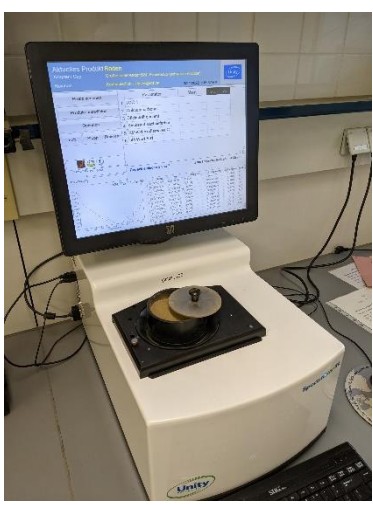

**Figure 2: SpectraStar<sup>TM</sup> XL near-infrared spectrometer from Unity Scientific loaded with a soil sample.**

The accompanying UCal™ Chemometric Software was used to pre-process the spectra. The reflectance ($R$) spectra were converted to absorbance ($A$) spectra by $A = \log_{10}(1/R)$ (Fig. 3). Scatter correction was achieved by standard normal variate (SNV) transformation and detrending (Barnes et al., 1989). The first forward derivative was applied to remove noise.

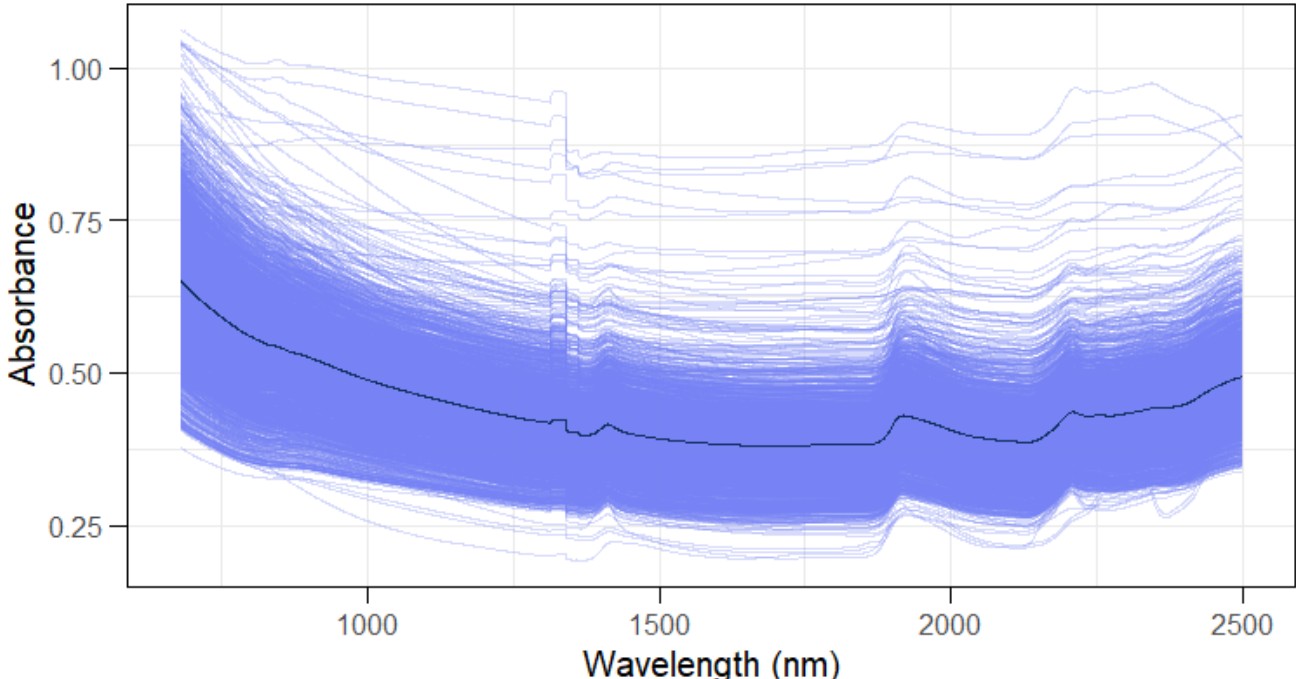

**Figure 3: Raw near-infrared soil spectra excluding standards (n = 2129) with mean spectrum (dark line).**





## 5 Spectroscopic modelling


The Kennard–Stone algorithm (Kennard and Stone, 1969) of the "prospectr" (Stevens and Ramirez Lopez, 2024) package in R Studio (version 2024.04.0) was used for calibration and validation sampling. The dataset was split into 80% calibration and 20% validation samples. To avoid pseudo-independence of spectra, grouped samples from the same sites were either assigned to the calibration or validation set via the algorithm. Simultaneously, a homogeneous distribution of samples was ensured by

applying the Mahalanobis distance. As these settings generated a very small validation set for clay in suspension (n = 7), a separate split using only samples with data of this soil property was conducted. Here, the Euclidean distance was used because the Mahalanobis distance requires more samples than individual spectra for point distance computation. The UCal™ Chemometric Software was then used to apply the partial least square regression (PLSR) for model calibration and validation. A maximum of 15 factors were set, and outlier detection and removal were performed by calculating the Mahalanobis distance,

which was set to a maximum global distance of 13 and 3 for the calibration and validation sets, respectively. The spectra were trimmed to the range of 780–2400 nm and sensor switches between 1330-1350 nm were removed. After model calibration, predictions were generated based on the validation set for each soil property. Models were tested for all samples or for subsets, i.e., SOC values below 7%, as this is the upper limit for most agricultural mineral soils in Austria. Statistical measures used to determine the performance of the model were the standard error of prediction (SEP) and the coefficient of determination ($R^2$)

from linear regression.

## 6 Model performance

NIR estimates of the 14 soil health indicators (Table 2 and Figure 4) revealed that PLSR explained total nitrogen contents (TN) best ($R^2$ = 0.9, SEP = 0.05), followed by $CaCO_3$ ($R^2$ = 0.89, SEP = 3.95), soil organic carbon (SOC) ($R^2$ = 0.82, SEP = 0.45) and clay ($R^2$ = 0.74, SEP = 4.35). Nonetheless, the standard error of prediction was large for these properties. This is a common

issue in larger-scale spectral libraries because they can comprise a wide range of soils with high mineralogical variability (Nocita et al., 2015; Stenberg et al., 2010), as it is the case for Austria. Regarding SOC, the regression shows that up to concentrations of 4%, the model predicted SOC well, but with increasing concentrations it underestimated NIR predictions due to the limited number of samples. Such limited numbers of samples for "extreme" values impacted the predictive quality of several other soil parameters, particularly phosphorus or sand content. Although SOC and clay results were sufficient, the

SOC-to-clay ratio was not well estimated by the model ($R^2$ = 0.49, SEP = 0.08); only narrow ratios showed good prediction by NIR spectroscopy. For total carbon (TC), only a small sample size for validation (n = 30) was available, which nevertheless revealed potential for accurate prediction by NIR ($R^2$ = 0.68, SEP = 1.02). Labile C had few low and high reference values, which impacted the prediction on NIR negatively ($R^2$ = 0.54, SEP = 83.79). When comparing the results for soil pH, pH in acetate achieved better estimates than pH in $CaCl_2$ ($R^2$ = 0.67, SEP = 0.2 and $R^2$ = 0.61, SEP = 0.61, respectively). Phosphorus

values above 200 mg kg$^{-1}$ were underestimated by NIR, yielding overall poor model results ($R^2$ = 0.2, SEP = 82.84). Clay



analyzed in suspension had a small coefficient of determination ($R^2 = 0.58$, SEP = 2.71). Other soil properties, i.e., cation exchange capacity (CEC), sand and silt, achieved $R^2$ below 0.5.





**Table 2: Validated NIR estimates (780–2400 nm) of 14 soil properties derived from PLSR. SOC is soil organic carbon, TC is total carbon, TN is total nitrogen, CEC is the cation exchange capacity, SEP is the standard error of prediction and GD is the global distance.**

| | SOC (%) | SOC/clay ratio | TC (%) | Labile C (mg kg⁻¹) | CaCO₃ (%) | TN (%) | Phosphorus (mg kg⁻¹) | pH CaCl₂ | pH Acetate | CEC (cmol kg⁻¹) |
|---|---|---|---|---|---|---|---|---|---|---|
| Training strategy | SOC <7 | SOC <7 | SOC <7 | All samples | All samples | All samples | All samples | All samples | All samples | All samples |
| $n$ initial | 405 | 160 | 30 | 161 | 89 | 247 | 315 | 424 | 124 | 173 |
| $n$ kept | 367 | 118 | 30 | 160 | 78 | 238 | 304 | 411 | 123 | 166 |
| Number of factors | 9 | 12 | 2 | 8 | 5 | 9 | 5 | 6 | 7 | 3 |
| Range of reference values | 0.15–6.85 | 0.03–0.78 | 1.54–6.17 | 87–1188 | 0.1–33.7 | 0.04–0.76 | 3–674 | 3.98–7.87 | 5.89–7.41 | 6.67–43.46 |
| SEP | 0.45 | 0.08 | 1.02 | 83.79 | 3.95 | 0.05 | 82.84 | 0.61 | 0.20 | 4.65 |
| R² | 0.82 | 0.49 | 0.68 | 0.54 | 0.89 | 0.90 | 0.20 | 0.61 | 0.67 | 0.40 |
| Mean GD | 1.17 | 0.58 | 0.32 | 0.75 | 0.95 | 0.96 | 1.16 | 0.99 | 0.60 | 0.42 |

| | Sand (%) | Silt (%) | Clay (%) | Clay in suspension (%) |
|---|---|---|---|---|
| Training strategy | All samples | All samples | All samples | All samples |
| $n$ initial | 178 | 178 | 166 | 76 |
| $n$ kept | 176 | 176 | 164 | 75 |
| Number of factors | 6 | 7 | 6 | 8 |
| Range | 7.7–68.7 | 20.8–5.7 | 2.8–47.1 | 14–36 |
| SEP | 11.06 | 9.82 | 4.35 | 2.71 |
| R² | 0.40 | 0.41 | 0.74 | 0.58 |
| Mean GD | 0.72 | 0.75 | 0.75 | 0.75 |





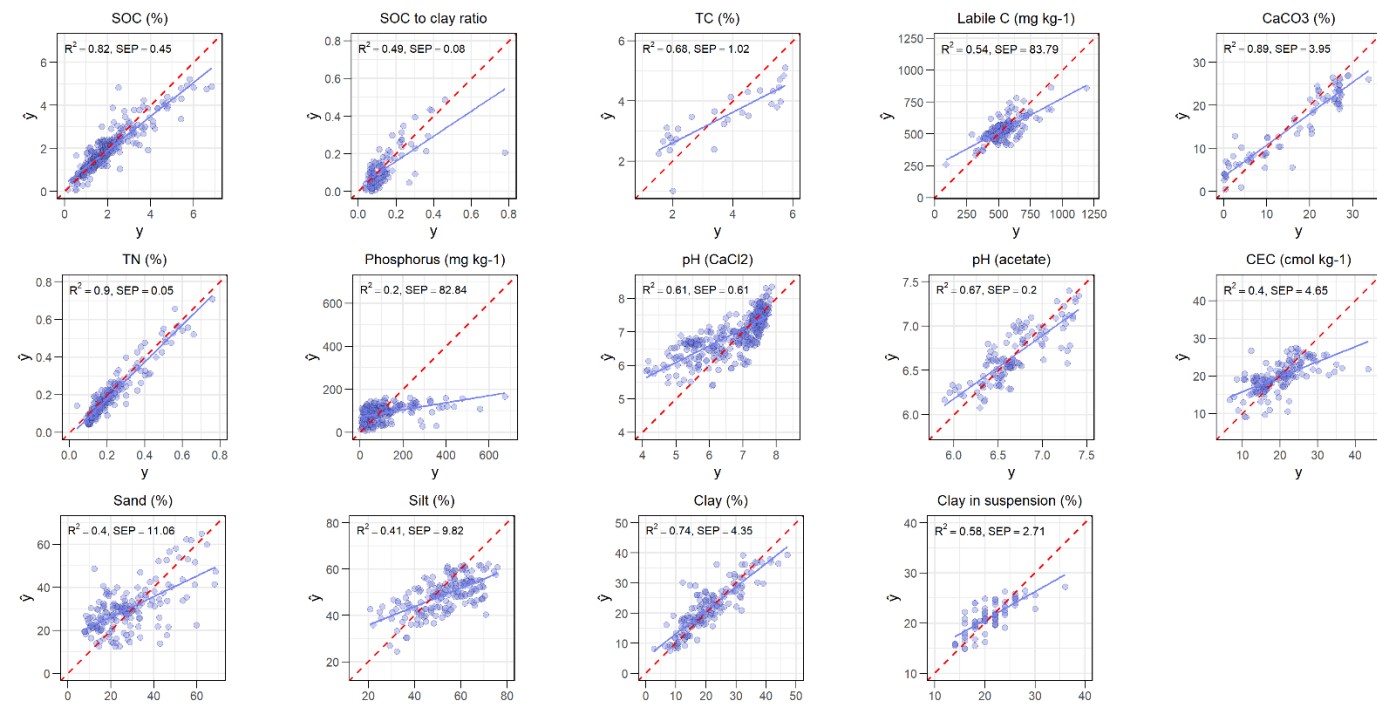

**Figure 4: Validation of the observed (y) versus predicted (ŷ) soil properties of the soil spectral library. SOC is soil organic carbon, TC is total carbon, TN is total nitrogen, CEC is the cation exchange capacity and SEP is the standard error of prediction. Blue line is the fitted line, while the red dashed line shows a regression of 1.**

## 7 Usability of the Austrian NIR Soil Spectral Library

The results of data validation show that applying partial least square regression to the Austrian NIR Soil Spectral Library yields estimates of TN, CaCO$_3$, SOC and clay contents with medium, and other properties with low accuracy. Although several properties were predicted with an $R^2 > 0.7$, standard errors of prediction were generally high. Considering the applicability for routine soil health analyses and monitoring, the predictive quality is currently insufficient compared to routine laboratory analyses. Nevertheless, in cases where funding is limited or rough estimates are sufficient, the tested models could be of benefit. Other user-cases might include the validation of citizen science data or the estimation of soil texture, as laboratory analysis is a tedious process. Increasing the sample number, particularly in the Alpine South, and including samples with "extreme" values, could improve the predictions and reduce standard errors. Grouping by environmental zones or according to the parent material are further options to assess for model improvement. Testing the performance of other models, particularly more advanced machine learning algorithms such as random forest, cubist, artificial neural networks or support vector regression (Viscarra Rossel et al., 2016; Minasny et al., 2024; Minasny and Mcbratney, 2008), could further enhance

the useability of the Austrian NIR Soil Spectral Library. Lastly, we encourage merging with other SSLs, for example by using transfer functions (e.g., calibration transfer (Feudale et al., 2002), harmonization functions (Francos et al., 2023)) or spiking for adaptation of models to the characteristics of the target sites (Guerrero et al., 2016) and using them for developing localized

calibration sets for specified contexts and pedologic domains (Viscarra Rossel et al., 2022).

## 8 Data availability

The dataset described in this manuscript can be accessed at the repository https://zenodo.org/ under the DOI 10.5281/zenodo.16261617 (Fohrafellner et al., 2025) and can be cited as follows: Fohrafellner, J., Lippl, M., Bajraktarevic, A., Baumgarten, A., Spiegel, H., Körner, R., and Sandén, T.: Dataset to: Austrian NIR Soil Spectral Library for Soil Health

Assessments (Version 3) [dataset], 10.5281/zenodo.16261617, 2025.

## 9 Conclusions

We present a first Austrian near-infrared (NIR) Soil Spectral Library with over 2100 legacy samples and test the performance of the partial least square regression (PLSR) to predict 14 chemical and physical soil health indicators. Several properties, i.e., total nitrogen, soil organic carbon, $CaCO_3$ and clay were estimated with medium accuracy, whereas the potential for predicting

other indicators varied. We conclude that applying the PLSR to the Austrian NIR Soil Spectral Library is not suitable for precisely estimating soil properties, but that the library presents a valuable foundation for future soil health assessments. Enlarging the spectral library would no doubt improve the predictions, specifically by including extreme values outside of common ranges and by including data from the Alpine South. Moreover, testing novel models using machine learning algorithms to train and validate the spectral library, for example, could improve predictions of soil health indicators. We

therefore encourage the use of this open dataset and of merging the spectra with other existing or forthcoming libraries. This effort is an important step forward in supporting the expansion of soil spectral libraries globally, facilitating the amplification of NIR analysis as a fast and simple method to assess, monitor and map soil health.

## Author contributions

JF and TS conceptualized and developed the methodology of this paper. JF and AB curated the data. JF conducted formal

analysis, developed the model code and visualizations and wrote the original draft. ML conducted formal analysis and validation. RK investigated reference samples. TS acquired funding and administered the project. All authors contributed to writing, reviewing and editing the manuscript.



**Competing interests**

The authors declare that they have no conflict of interest.

**Acknowledgements**

This research was conducted within the ProbeField project, which was part of the European Joint Program for SOIL 'Towards climate-smart sustainable management of agricultural soils' (EJP SOIL) funded by the European Union Horizon 2020 research and innovation programme (Grant Agreement N° 862695). We further would like to thank Georg Dersch for planning the sample selection, Michael Schwarz for creating the Austrian map and assisting in data extraction, as well as Günther Aust,
Hans-Peter Haslmayr, Florian Forcher and Monika Tulipan for providing meta-data on the soil samples. Moreover, we want to thank Bo Stenberg and Fabio Castaldi for providing advice on this manuscript. Michael Stachowitsch helped with language editing.



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
