# Peer review of "Austrian NIR Soil Spectral Library for Soil Health Assessments"

_Earth System Science Data, 2025_

## Author Response (AR1)

**Response to reviewer comments**

**RC2**: 'Comment on essd-2025-426', José Lucas Safanelli, 01 Nov 2025

The paper "Austrian NIR Soil Spectral Library for Soil Health Assessments" introduces the first openly-accessible Austrian Near-Infrared (NIR) soil spectral library and explores its potential for soil health assessment via spectroscopy. The paper is well-structured and well-written, and I don't have any major objections to its publication. In fact, I congratulate the authors on their effort, as the soil spectroscopy and Austrian soil science communities can greatly benefit from this release.

**Response:**

Thank you very much for taking the time to review our manuscript, for the positive feedback and for providing highly useful comments. We have incorporated your suggestions to the best of our ability.

A few comments:

C1:
Dataset: Wonder if it would be possible to release the spatial coordinates as presented in Figure 1. If, for some reason, a site cannot be disclosed (e.g., privacy concerns), you can downgrade the coordinate up to 2 decimal points. If possible, provide position accuracy (and datum) to help filter sites: those downgraded points would have a precision of 1km. In contrast, sites with 5-6 decimal points in their coordinates would have a precision ranging within meters. The authors and other scientists can explore the integration of NIR with spatial data layers or metadata to enhance performance with a locally contextualized model.

**Response:**
Thank you for this suggestion, which in case coordinates were available would have been our first choice. I did check the raw data again, but unfortunately for the majority of sites no coordinates but only village names or zip codes are available. This can be explained by the fact that i) the AGES soil laboratory is the key facility in Austria for analyzing soil samples sent in by practitioners, who do not indicate the coordinates of sample retrieval. These samples do give us the opportunity to receive a large distribution throughout Austria for our NIRS library, as Austria so far does not have an ongoing monitoring scheme. Nevertheless, this comes with several disadvantages, as missing coordinates; ii) The incorporation of "soil box" samples from private individuals (not only farmers), who also are not asked to disclose any coordinates; iii) Inclusion of samples from taken by external research facilities, for which accessibility to coordinates was not always possible.

Nevertheless, we completely agree with your suggestions and therefore decided to include additional information on the issue of missing coordinates in section 3 Dataset creation and description (L105-108) and also added a sentence on the limitations of our dataset in 7 Usability of the Austrian NIR Soil Spectral Library (L242-243).

C2:

Spectral measurement and preprocessing: Looking at the spectral variation, I wonder if the authors have corrected the splice/bump around 1300-1400 nm, as this may create artifact features when rolling a smoothing or first derivative window across the spectra. The Prospectr package from R has a function to correct that. Just need to indicate the splice wavelength precisely.

**Response:**

We did remove the sensor switch between 1330-1350 nm which is visible in Figure 3 (L180-181). Besides SNV, detrending and first derivative, no other modifications were conducted, as the UCal™ Chemometric Software did not detect issues with the processed spectra.

C3:

Spectroscopic modeling: kudos for ensuring that soil samples were grouped according to their site location. In Line 165, can you better elaborate on how a homogeneous distribution of samples was ensured with Mahalanobis or Euclidean distances?

**Response:**

Thank you very much! We changed the term "homogenous" to "uniform", as used by Stevens and Ramirez Lopez (2024), and added several sentences (L168-169, L172-173, L175).

C4:

Line 175. It would be important to provide equations for SEP and $R^2$, especially because $R^2$ can be calculated in multiple ways. Would it be possible to add RPIQ (IQR/RMSE, or some standardized SEP [SD/SEP or IQR/SEP] that is not affected by the scale and range of soil properties) to allow a better comparison among soil properties? Would it be good to provide additional metrics like RMSE and bias? The soil spec community is more familiar with these metrics.

**Response:**

We went through the UCal™ Chemometric Software manual to determine the equations for computing SEP and $R^2$, which are unfortunately not provided. After contacting UCal™

support, we learned that the equations are not documented and that the developer responsible for the software is no longer employed. Unfortunately, this leaves us unable to provide the exact formulas.

Nevertheless, we have now included RPIQ in Table 2 to enable better comparison among soil properties. The inclusion of other metrics is not possible, as the software only calculates the metrics listed in Table 2.

**RC1**: 'Comment on essd-2025-426', Anonymous Referee #1, 13 Oct 2025 reply

**General Comments**

This manuscript presents a valuable and timely contribution to the field of soil science and digital soil mapping. The development of the first open-access Austrian NIR soil spectral library (SSL) fills a significant data gap and aligns perfectly with current European initiatives (e.g., EU Soil Mission, Soil Monitoring Law) that demand cost-effective tools for monitoring soil health. The study is well-structured, the methodology is sound and thoroughly described, and the data is made openly available, which is highly commendable. While the current predictive performance of the PLSR models for most properties is reported as insufficient for replacing routine lab analyses, the library itself represents a crucial foundational resource for the scientific community. The manuscript is therefore suitable for publication in Earth System Science Data after minor revisions to clarify certain aspects and strengthen the discussion.

**Response:**

We greatly appreciate the reviewer's time and detailed assessment of our manuscript. We trust that the revisions and responses provided adequately address your comments.

**Specific Comments**

**Abstract and Short Summary:**

**L13-15 (Short Summary):** The statement "the accuracy was insufficient compared to routine laboratory analyses" is very general. Consider rephrasing to be more specific and balanced, e.g., "The accuracy for most properties was currently insufficient... though several key properties (TN, SOC, $CaCO_3$, clay) showed promising predictive potential ($R^2 > 0.7$)."

**Response:**

The short summary has a limit of 500 characters including spaces, so adding text was difficult. We changed the second sentence in a similar way you suggested, but due to character limitations shortened it.

**L28-30 (Abstract):** Similar to above. The phrase "is not suitable to predict most of the 14 soil properties with sufficient accuracy" could be tempered to "showed limited accuracy for predicting many of the 14 soil properties", followed immediately by the positive results for TN, etc.

**Response:**

Thank you, we adapted the sentences.

**Introduction:**

**L53-55:** The sentence "Based on the increasing requirements... are in demand" is a bit awkward. Suggest rephrasing for clarity: "The increasing requirements for soil health assessments... are creating a demand for less cost-intensive alternative methods."

**Response:**

Thank you, we made changes as suggested.

**Soil sample selection:**

**L89-90:** "For one sample, the location and environmental zone are unknown (Sample_number 743)." It is good practice to state how this sample was handled in the spatial analysis (e.g., was it excluded from Fig. 1?). Please clarify.

**Response:**

We added in L89 that the sample is not included in Figure 1 and modified the structure of the sentences to clarify.

**Figure 1:** The figure is essential. Please ensure that in the final version, the map is of high resolution and the circle sizes for sample counts are clearly distinguishable in the legend.

**Response:**

Inserting it in the word was sub-optimal, as it cannot be read clearly there. We will ensure that it will be bigger and in higher resolution in the final publication.

**Dataset creation and description:**

**L108-110:** "Providing coordinates was not possible because the dataset includes samples sent in by private individuals..." This is a crucial point regarding data FAIRness (Findability). It is well-justified, but it should be explicitly mentioned in the "Data availability" section as a limitation of the dataset's interoperability.

**Response:**

Upon investigating the manuscript guidelines, we believe that in the Data Availability section no additional information besides the source of the dataset

should be stated. The second reviewer also commented on this issue, so we added additional sentences in L106-108 and L242-243. We hope you feel this is adequate.

**L111-112:** "Sampling depths are reported in columns 8 and 9". Please specify what these two columns represent (e.g., "upper depth" and "lower depth" or "min depth" and "max depth"?).

**Response:**

Thank you, we made changes as suggested.

**Chemical and physical reference analysis:**

**Table 1:** The minimum and maximum values for silt content (5% and 75.7%, respectively, in Table 1) seem unusual given the range of the clay fraction. Could the authors please double-check these values for potential typographical errors?

**Response:**

Thank you, we double checked the texture minimum and maximum values in the dataset, and they are in fact correct. The samples in our dataset do cover a truly wide range!

**L142-146:** The paragraph explaining the SOC < 7% subset is critical for understanding the modelling choices. This rationale should be briefly restated in Section 5 ("Spectroscopic modelling") when the models are introduced, as it is key to interpreting the results in Table 2.

**Response:**

We now added sentences to section 5, where we refer to section 4.1 and explain again, why the SOC < 7% subset was necessary.

**Spectral measurement and preprocessing:**

**L159:** "The first forward derivative was applied to remove noise." Derivatives are typically used to enhance spectral features and remove baseline offsets; noise removal is usually achieved by smoothing. Please clarify the intended purpose here.

**Response:**

Thank you for the correction, we now added the purpose of removing baseline shifts and added a reference.

**Spectroscopic modelling:**

**L175-178:** The explanation for handling the "clay in suspension" validation set is clear and logical.

**Response:**

Thank you!

**L179:** Please specify the version of the prospectr package used for reproducibility.

**Response:**

The prospectr package version (0.2.7) was added.

**L182:** "UCal™ Chemometric Software". If this is a commercial software, please provide the company and location (e.g., Unity Scientific, MA, USA) for completeness.

**Response:**

We added the company name and location to L160 and L177.

**Model performance & Figure 4:**

**L200-201:** "clay analyzed in suspension had a small coefficient of determination ($R^2$=0.58, SEP=2.71)". An $R^2$ of 0.58 is actually quite respectable for soil spectroscopy, especially for a physical property. Consider using a more neutral term like "moderate" instead of "small".

**Response:**

We made the change as suggested.

**Figure 4:** The plots are excellent and very informative. Please ensure all axis labels are clearly visible in the final version. The unit for Labile C is cut off in the provided preprint (mg kg$^{-1}$).

**Response:**

Thank you very much! We will upload the figure separately to ensure high resolution and visibility of all labels in the final submission process.

**Usability of the Austrian NIR Soil Spectral Library:**

**L222-223:** "the predictive quality is currently insufficient compared to routine laboratory analyses." This is a key conclusion. It would be helpful to provide a specific threshold or benchmark the authors have in mind for "sufficient" accuracy (e.g., RPD > 2, or a required SEP for practical application).

**Response:**

We now added the ratio of performance to inter-quartile distance (RPIQ) (Bellon-Maurel et al., 2010) and chose a threshold of 1.89 (Ludwig et al., 2019) (see Table 2 and L204-213). A maximum SEP across all parameters cannot be set, as the SEP is linked to the individual units and not standardized across properties, as the RPIQ. Nevertheless, we focused our model performance assessment on R2 and SEP, as SEP (precision) is an important indicator when thinking of application of NIR instead of routine laboratory measurements.

**L230-235:** The suggestions for improvement are excellent. To make this section even stronger, consider structuring it into a short, bulleted list or a separate paragraph titled "Recommendations for Future Work".

**Response:**

Thank you very much! To avoid having only little flow text in section 7 (as more than half of the text are recommendations) we prefer to have this paragraph as one flow text. Nevertheless, we changed the section title to "7 Usability of the Austrian NIR Soil Spectral Library and recommendations" to make this more visible.

**Data availability:**

As mentioned above, please add a note here about the lack of precise coordinates due to privacy concerns, acknowledging this as a limitation for certain spatial applications.

**Response:**

As mentioned above, we added additional sentences on this weakness in L106-108 and L242-243.